# The Effect of Addition of Antioxidant 1,2-dihydro-2,2,4-trimethyl-quinoline on Characteristics of Crepe Rubber Modified Asphalt in Short Term Aging and Long Term Aging Conditions

**Bahruddin Ibrahim [1],\***  **, Arya Wiranata [1] and Alfian Malik [2]**

1   Chemical Engineering Department, Engineering Faculty, University of Riau, Pekanbaru 28923, Indonesia; arya.wiranata7066@grad.unri.ac.id

2   Civil Engineering Department, Engineering Faculty, University of Riau, Pekanbaru 28923, Indonesia; ALFmalik@gmail.com

\*   Correspondence: bahruddin@lecturer.unri.ac.id

**Abstract:** The use of natural rubber to resist bitumen is usually prone to degradation and aging. One method to overcome this problem is the addition of antioxidants. This study aims to determine the effect of the addition of antioxidants 1,2-dihydro-2,2,4-trimethyl-quinoline (TMQ) on natural rubber modified asphalt crepe rubber and its performance in short-term and long-term aging conditions. The modified rubber asphalt mixture's manufacture begins by melting the crepe rubber at 200 °C before being mixed in asphalt at 165 °C. Addition of antioxidant TMQ during the stirring process of the mixture of asphalt and melted rubber. The crepe rubber content was 8, 10, and 12% *w/w*, while the TMQ content was 1, 2, and 3% *w/w* of the total sample weight. The modified asphalt samples' characterization included penetration, softening point, weight loss after a rolling thin film oven test (RTFOT), penetration after RTFOT, and Marshall test. Review of the performance of asphalt under short-term aging conditions using a dynamic shear rheometer (DSR). Evaluation of asphalt performance under long-term aging conditions using Fourier-transform infrared spectroscopy (FTIR). The results showed that the fact that the best-modified asphalt product was the addition of 10% crepe rubber and 2% TMQ. The best-modified asphalt characteristics have penetration 68.70 dmm, softening point 55.45 °C, weight loss only 0.0579%, penetration after RTFOT 59.60, Marshall stability 1403.96 kg with optimum asphalt content of 5.50%, and rutting factor (G*/Sinδ) 6.91 kPa and 16.1 kPa before and after RTFOT. Overall, the modified crepe rubber asphalt can improve the performance of the asphalt in terms of durability. Simultaneously, the antioxidant TMQ works very well in increasing the resistance of bitumen to aging in the conditions of short-term aging and long-term aging.

**Keywords:** asphalt rubber; crepe rubber; Marshall stability; rolling thin film oven test; 1,2-dihydro-2,2,4-trimethyl-quinoline

## 1. Introduction

The uncontrolled growth of land transportation in many developing countries causes many problems, such as traffic jams due to bad roads to accommodate the number of vehicles. One of the developing countries with a high level of congestion is Indonesia. Traffic congestion with a long tempo puts a significant burden on road construction and damages the pavement structure of asphalt roads. Other factors that support the damage to asphalt roads are premature aging caused by standing water, humidity, heat, UV, oxidation, and low asphalt quality [1,2]. This results in higher road maintenance costs. The solution to reducing the high maintenance costs of paved roads is to review road construction

and improve asphalt quality through a modification process. The use of natural rubber modified asphalt can produce more resistant to high traffic loads and prevent the asphalt from aging prematurely.

Asphalt modification by adding additives is a common practice to improve the quality of the asphalt binder. The additive criteria for asphalt mixtures must produce stability and high softening points, increase flexibility, increase durability, increase the bonding power of asphalt to aggregates, and prevent the asphalt's premature aging. Indonesia has many options for additives mixed in asphalt, such as plastic waste, styrofoam waste, used tire waste, natural rubber, etc. Natural rubber is the most potential material as an additive to the asphalt mixture from the many available additives. The natural rubber has advantages in lower rubber asphalt production costs, very large quantities of available and better quality than crumb rubber from used tires. It does not cause emissions after being used as an asphalt additive.

The natural rubber used can be latex, solid rubber, or engineering rubber. The development of natural rubber in other sectors is a must, such as an additive for asphalt modification. The development of natural rubber as an additive to modified asphalt must be carried out immediately, especially for several natural rubber producing countries that have experienced a decline in natural rubber prices. The growing use of natural rubber for modified asphalt is a solution to the decline in natural rubber prices in recent years.

Several studies have conducted using crumb rubber (CR) from used tires as an additive to asphalt mixtures. They use CR from desulphurization, and non-desulphurization used tires with a size of 40 mesh as a mixture on 90 dmm penetration asphalt. CR is mixed as much as 20% at a mixing temperature of 180–190 °C with a stirring time of 1 h. The test results showed that the addition of CR without desulphurization increased viscosity and storage stability compared to desulphurization. However, the use of CR desulphurization provides several disadvantages, including higher density and high mixing temperature, poor storage stability, and requires large amounts of bitumen for field applications [3].

The addition of 20% *w/w* Crumb rubber (CR) from used tire rubber with desulphurization treatment and a 40 mesh size positively impacts modified asphalt's rheological properties. The desulphurization process makes CR dissolve more easily in asphalt compared to mixing CR directly into the asphalt. However, CR directly has a higher softening point than asphalt with a desulphurization CR mixture. The addition of CR by desulphurization treatment showed lower viscosity and better storage stability when compared to adding CR directly in asphalt. Asphalt, with the addition of CR desulphurization, requires a higher production temperature, which is the main drawback of this process [4].

The use of 4% *w/w* styrene butadiene styrene (SBS) and 20% *w/w* CR from used tires with a size of 40 mesh and 80–10 mesh has a positive effect on modified asphalt rheology. The addition of CR results in better resistance to permanent deformation, rutting resistance, anti-fatigue, crack resistance, moisture susceptibility, and better storage stability than modified bitumen with SBS. However, CR mixing requires a high temperature and stirring speed compared to asphalt mixtures with SBS [5]. CR with a size of 30 mesh produces modified asphalt that is more stable. CR and asphalt were mixed at 170 °C with a stirring time of 1 h. The test results show that the addition of CR with a size of 30 mesh has better performance at low to moderate temperatures but less good on storage stability [6].

Apart from CR from used tires as an asphalt additive, natural rubber (NR) is another alternative additive for asphalt modification. The rubber used can be latex, cup lump (CL), technical rubber (such as crumb rubber), or used rubber from used tires. The use of latex with a dry rubber content >60% recommendation for asphalt modification additives. This latex can reduce the sensitivity of bitumen to temperature, increase resistance during use, and even distribute rubber in the mixture. Still, this process requires a large amount of latex and a long pre-treatment [7]. CL's addition increased the asphalt's softening point but significantly decreased its penetration and ductility [8]. Use of block skim rubber (BSR) and crumb rubber standard Indonesian rubber (SIR 20), cemented in open mills. This mastication treatment can reduce the Mooney viscosity, speed up mixing time, and increase the

asphalt's softening point [9]. The use of activated rubber from used tires as an asphalt additive also increased asphalt softening points [10].

The polymer as an asphalt additive has several disadvantages, including degradation, oxidation, and susceptibility to free radicals in rubber, which accelerate asphalt aging. Many studies with various other additives have reported minimizing the aging of rubber modified asphalt [2]. The addition of 4–6% *w/w* nano-silica and 4% *w/w* diatomite gives an effect as an oxidation inhibitor. It protects the bitumen from oxygen, but the increased cohesion after adding nano-silica and diatomite makes the asphalt susceptible to rutting and fatigue [11]. The use of 1–2% *w/w* Irganox 1010 and Dilauryl Thiodipropionate (DLTP) tends to make the asphalt binder soft so that the asphalt sensitivity to low temperatures decreases, and the asphalt is prone to rutting [12]. Many additives have been used and studied to inhibit the oxidation of polymer asphalt mixtures. Still, the addition of other additives has an impact on reducing the mechanical performance of polymer bitumen.

The use of solid rubber, such as crumb rubber, will take a very long time in the mixing process, so a depolymerization process by chewing is required [9]. Mastication is the process of changing the physical properties of rubber from elastic to plastic. During mastication, the finished polymer chain breaking is due to shear forces by the rolling mill. This process results in a decrease in the molecular weight and viscosity of the rubber. The most efficient mastication process is carried out at temperatures below 60–70 °C and above 120–130 °C [13]. The mastication process can take place faster by adding a plasticizer as a rubber softener. Asphalt is used as a plasticizer to save rubber asphalt production costs. Compounds derived from hydrocarbon oil are very suitable for use as plasticizers for natural rubber types. Asphalt is a hydrocarbon oil compound, so it is ideal for plasticizers for natural rubber [14].

Based on these studies, there are several conclusions that the modification of asphalt using various types of natural rubber has a positive effect on improving the quality of asphalt. But the use of natural rubber is also susceptible to degradation, oxidation, and the formation of free radicals, which accelerate the premature aging of the asphalt. Based on this study, this study aims to make asphalt with high durability, resistance to low or moderate temperatures, resistance to rutting, and to prevent premature aging of asphalt. This study used solid natural rubber crepe rubber with the addition of antioxidants 1,2-dihydro-2,2,4-trimethyl-quinoline (TMQ). This research will study the effect of adding crepe rubber and antioxidant TMQ on modified asphalt characteristics.

## 2. Material and Method

### 2.1. Material

This study uses the main ingredient, namely, Asphalt Penetration Grade (Pen Grade) 60/70, with the specifications according to Table 1, PT's production. Pertamina (Persero). The additive used for this study uses natural rubber, which is processed by itself from the raw ingredients of cup lump to crepe rubber. The cup lump used came from rubber plantations in Kampar Regency, Riau Province, and other additives used were Antioxidant TMQ. Aggregates used in this study use processed aggregates from PT. Virajaya Riau Putra, Riau Province. The aggregate specifications used can be seen in Table 2.

**Table 1.** Specifications for asphalt penetration grade (pen grade) 60/70.

| Parameter | Test Standard | Result |
|---|---|---|
| Penetration at 25 °C (dmm) | ASTM D5 | 70.2 |
| Softening point (°C) | ASTM D36 | 48 |
| Weight loss by Thin Film Oven Test (TFOT) (%) | ASTM D6 | 0.365 |
| Penetration after Thin Film Oven Test (TFOT) (dmm) | ASTM D5 | 64.7 |
| Ductility (cm) | ASTM D113 | 110 |
| Marshall stability (kg) | ASTM D6927 | 1179.25 |

**Table 2.** Aggregate composition of Marshall stability testing.

| Aggregate | Composition (%) |
|---|---|
| Coarse aggregate | 15 |
| Medium aggregate | 30 |
| Fine aggregate | 53 |
| Filler cement | 2 |

*2.2. Method*

A.    Preparation of Crepe Rubber

The first is to wash the cup lump in running water to remove the dirt that sticks to the cup's lump surface. Cup lump milled using a creeper machine to reduce the moisture content of the cup lump. During the grinding process, the cup lump is flowed with water to remove dirt that did not come off during the first washing process. After the cup lump is sheet-shaped, the cup lump sheet is cut into pieces and washed again to remove dirt inside the rubber. Clean cup lumps are placed in the open roll mill machine in cold operating conditions to form crepe rubber sheets. The crepe rubber sheet is dried in an open space without being exposed to sunlight for 7–14 days for the drying process to obtain dry rubber content (DRC) > 95%.

B.    Making Modified Asphalt Samples

Modified asphalt samples begin with weighing the asphalt, crepe rubber, and TMQ antioxidants with the ratio, as shown in Table 3. First, cut a sheet of crepe rubber with a size of 1 cm × 1 cm. Next, melt the crepe rubber pieces at 200 °C and add asphalt with a 1:1 ratio to speed up the melting process. Asphalt and crepe rubber dissolved were mixed by melting method at 165 °C with stirring at 300 rpm for 30 min. The asphalt sample was then rested before being tested for the modified asphalt characterization and the sample code shown in Table 3.

**Table 3.** Modified asphalt sample ratio.

| Sample Asphalt | Content (%) | | |
|---|---|---|---|
| | Asphalt | Crepe Rubber | TMQ |
| Pen Grade 60/70 | 100 | 0 | 0 |
| A1 | 91 | 8 | 1 |
| A2 | 90 | 8 | 2 |
| A3 | 89 | 8 | 3 |
| B1 | 89 | 10 | 1 |
| B2 | 88 | 10 | 2 |
| B3 | 87 | 10 | 3 |
| C1 | 87 | 12 | 1 |
| C2 | 86 | 12 | 2 |
| C3 | 85 | 12 | 3 |

C.    Aging Method

The asphalt aging process takes a very long time. Simulation is a solution to speed up the aging process of asphalt to get actual results. Simulations to obtain short-term aging data used the heating treatment on asphalt using the rolling thin film oven test (RTFOT) with ASTM D2872. Meanwhile, long-term aging is simulated by heating the modified asphalt sample after RTFOT utilizing an oven. The oven is set at 85 °C with a heating time of eight days [15].

*2.3. Characterization of Modified Asphalt Samples*

The characterization of asphalt-rubber mixture samples included penetration testing (ASTM D5), softening point (ASTM D36), weight loss (ASTM D6/D6M), penetration after TFOT (ASTM D5), and Marshall stability (ASTM D6927). Marshall stability testing uses the aggregate composition shown in Table 2 with the optimum bitumen content, as seen in Table 4. The modified asphalt characterization test results are shown in Table 4 and the Marshall stability test shown in Table 5. Furthermore, the dynamic shear rheometer (DSR) test describes asphalt's viscoelastic behavior after short-term aging using RTFOT. Evaluation of modified asphalt performance after the addition of crepe rubber and TMQ carried out DSR testing for asphalt samples of pen grade 60/70, asphalt with optimum crepe rubber without TMQ, and asphalt with the addition of crepe rubber and TMQ at optimum conditions. DSR testing is carried out on these samples before and after RTFOT (short term oven aging). The Fourier-transform infrared spectroscopy (FTIR) test was carried out to determine modified asphalt performance after long-term aging.

**Table 4.** Test results of modified asphalt characteristics.

| Sample | Optimum Asphalt (%) | Penetration (dmm) | Softening Point (°C) | Weight of Loss by RTFOT (%) | Penetration after RTOFT (dmm) | Decreased Penetration after TFOT (%) |
|---|---|---|---|---|---|---|
| Pen grade 60/70 | 6.00 | 70.20 | 48.00 | 0.365 | 64.70 | 7.83 |
| A1 | 5.45 | 61.30 | 58.70 | 0.066 | 50.80 | 17.13 |
| A2 | 5.45 | 63.10 | 57.15 | 0.096 | 53.50 | 15.21 |
| A3 | 5.45 | 71.60 | 52.65 | 0.096 | 56.40 | 21.23 |
| B1 | 5.50 | 64.70 | 55.80 | 0.063 | 45.90 | 29.06 |
| B2 | 5.50 | 68.70 | 55.45 | 0.059 | 59.60 | 13.25 |
| B3 | 5.50 | 74.40 | 53.65 | 0.063 | 53.00 | 28.76 |
| C1 | 5.35 | 66.20 | 55.80 | 0.125 | 54.20 | 18.13 |
| C2 | 5.35 | 69.70 | 55.60 | 0.065 | 62.50 | 10.33 |
| C3 | 5.35 | 78.30 | 47.65 | 0.125 | 57.60 | 26.43 |

**Table 5.** Characterization of Marshall stability.

| Sample | Optimum Asphalt (%) | Marshall Stability | VFA (%) | VIM (%) | VMA (%) | Flow (mm) | MQ (kg/mm) |
|---|---|---|---|---|---|---|---|
| A1 | 5.45 | 1336.24 | 68.737 | 4.706 | 15.082 | 3.370 | 396.510 |
| A2 | 5.45 | 1426.53 | 67.928 | 5.374 | 16.555 | 3.710 | 384.509 |
| A3 | 5.45 | 1363.33 | 68.739 | 4.973 | 15.903 | 3.900 | 349.572 |
| B1 | 5.50 | 1354.30 | 77.686 | 3.366 | 15.082 | 3.880 | 349.046 |
| B2 | 5.50 | 1403.96 | 75.907 | 3.071 | 15.340 | 3.370 | 416.605 |
| B3 | 5.50 | 1101.50 | 77.172 | 3.470 | 15.194 | 3.820 | 288.351 |
| C1 | 5.35 | 1074.41 | 70.106 | 4.369 | 14.654 | 3.630 | 295.981 |
| C2 | 5.35 | 1254.98 | 72.055 | 4.078 | 14.522 | 3.990 | 314.531 |
| C3 | 5.35 | 1110.53 | 70.604 | 4.296 | 14.624 | 4.370 | 254.126 |

## 3. Result and Discussion

### 3.1. Effects of Crepe Rubber and TMQ on Modified Asphalt Penetration

Penetration describes the physical hardness level of the asphalt and is a parameter for classifying the quality of asphalt. Asphalt with a lower penetration value is more suitable for use in areas with hot climates. Asphalt with low penetration values generally has higher softening points and rutting resistance, thus extending the asphalt life. Asphalt test results using an automatic penetrometer as presented in Table 4. Asphalt pen grade 60/70 has an asphalt penetration value of 70.2 dmm. Modified asphalt crepe rubber and antioxidant TMQ decreased and increased penetration of asphalt compared to pen grade 60/70 asphalt. Figure 1 shows that modified asphalt tends to increase penetration and the increasing ratio of crepe rubber and TMQ. The penetration values of Samples A1, A2, B1, B2, C1, and C2 as a whole are lower than asphalt pen grade 60/70, while samples A3, B3, and C3 have a higher penetration value than asphalt pen grade 60/70.

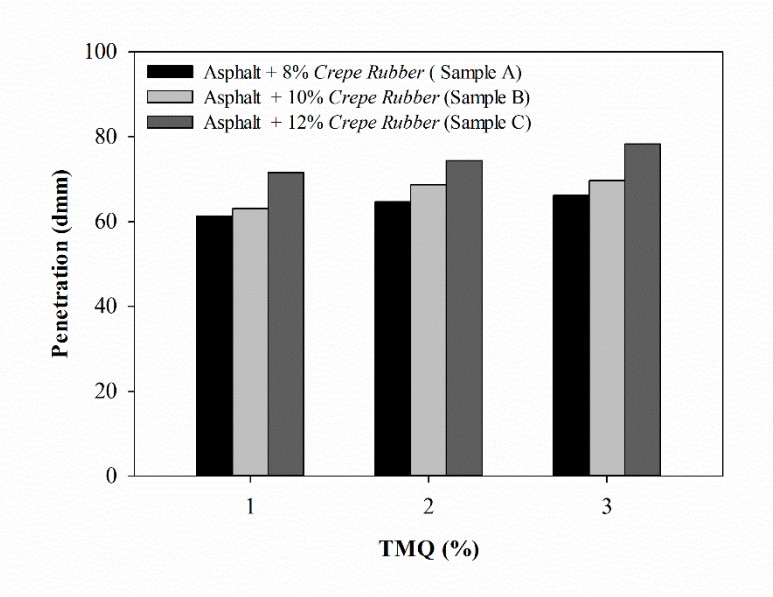

**Figure 1.** Penetration.

The lowest penetration value is in sample A1 with a penetration value of 61.30 dmm or a decrease of 12.67% from the 60/70 pen grade asphalt. The highest penetration is in the C3 sample with a penetration value of 78.30 dmm or an increase of 85.61% from the 60/70 pen grade asphalt. In general, the decrease in asphalt penetration is caused by an increase in the ratio of crepe rubber in asphalt,

increasing the absorption capacity of the light fraction of asphalt and accelerate the distribution of melted crepe rubber [16]. The absorption of light fractions of asphalt, such as saturate and some aromatic compounds by crepe rubber, will physically change the crepe rubber to expand and form a viscous gel [17].

An increase in the ratio of crepe rubber that exceeds the optimum will cause the asphalt to become soft. The melted crepe rubber is not evenly mixed in the asphalt and accumulates on the asphalt-rubber mixture. The accumulation of rubber changes the modified asphalt mixture's surface to be softer so that the penetration value increases during testing. An increase in the ratio of crepe rubber that exceeds the optimum will cause the asphalt to become soft. The melted crepe rubber is not evenly mixed in the asphalt and accumulates on the asphalt—rubber mixture. The accumulation of rubber changes the modified asphalt mixture's surface to be softer so that the penetration value increases during testing [8].

Several factors cause the accumulation of melted crepe rubber on the asphalt surface, including the slow stirring speed during the mixing process, which is the main cause of the accumulation of melted crepe rubber on the surface asphalt mixture-crepe rubber. During the mixing process of asphalt with crepe rubber, the stirring speed plays an important role in the distribution speed of melted crepe rubber in the asphalt [8,9]. The direct use of crepe rubber takes a very long time to melt. The solution to speed up crepe rubber's melting process is to use high temperatures, but this has drawbacks. The melting process of crepe rubber at 200 °C for a long time can cause degradation and damage the polymer structure of crepe rubber or asphalt. Degradation of structural damage to the crepe rubber is undesirable because it will affect the asphalt sample's physical properties to become softer [18].

Samples A1, A2, A3 has the same ratio but have a penetration value that increases with increasing levels of TMQ in the mixture. The mixing temperature in the range of 165 °C causes TMQ to melt and dissolve in asphalt because the melting point of TMQ is the only 72–94 °C. The physical properties of TMQ melt are different from that of crepe rubber melt. TMQ melt has a more fluid nature, which affects the modified asphalt mixture's consistency to become soft. TMQ has aromatic and aliphatic molecular structures so that TMQ melts have properties similar to maltene compounds in asphalt, which are mostly composed of aromatic and alpha molecular structures [19]. Overall, the increase in TMQ will increase the maltene component in the asphalt, which affects the asphalt's physical properties to soften. Increasing the TMQ ratio will limit the interaction of rubber in the asphalt so that the modified asphalt becomes softer [20].

### 3.2. Effects of Crepe Rubber and TMQ on Modified Asphalt Softening Points

Asphalt modification with crepe rubber was declared successful if the softening point value of modified asphalt was higher than the asphalt pen grade 60/70 soft point value [21]. The softening point increase occurred in almost all samples except C3. Samples A1 and A2 had the highest softening points with 58.70 °C and 57.15 °C, or an increase of 19–22% of asphalt pen grade 60/70. The lowest softening point in sample C3 with a softening point value of 47.65 °C is shown in Figure 2 and Table 4. The increase in softening point is proportional to the increase in asphaltene and semi-solid (resin) levels accompanied by a linear decrease in maltene content and decreased asphalt penetration value [22]. Asphaltene increases with increasing levels of crepe rubber in asphalt until it reaches its optimum level. Crepe rubber in asphalt will absorb maltene to form a viscous gel, strengthening the interaction of asphaltenes and resin.

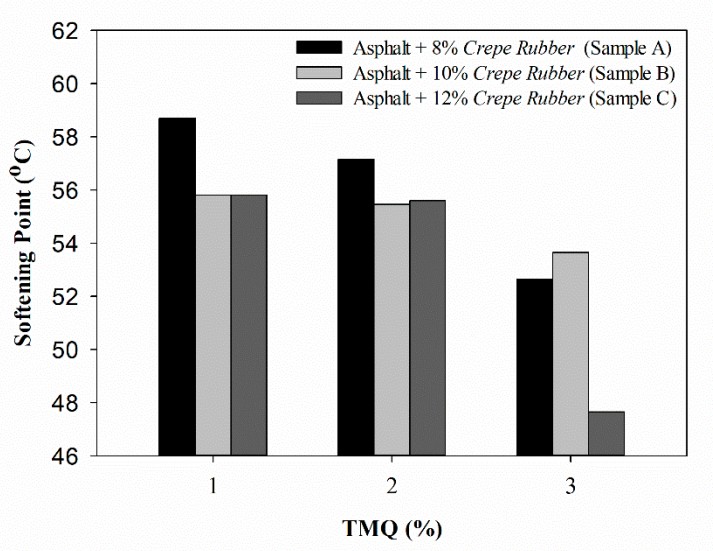

**Figure 2.** Softening point.

Asphalt's softening point decreases with increasing crepe rubber levels in modified asphalt, as shown in samples A1. B1. C1 with the same TMQ levels. The same is seen in samples A2, B2, C2, and A3, B3, C3. The softening point decreases inversely with the increase in asphalt penetration, caused by accumulated melt formation in the modified asphalt sample. In general. TMQ does not affect softening points directly but does affect asphalt penetration. TMQ only plays a role in preventing oxidization of rubber or asphalt during its lifetime and storage. Meanwhile, crepe rubber plays a significant role in increasing the softening point and rutting resistance at low temperatures when crepe rubber stabilizes in the swelling process [23]. Table 4 shows the soft points of all samples. B2, B3, C1, and C2 have almost the same soft points in the 55 °C temperature range and adjacent penetrations in the 64–70 dmm range.

Crepe rubber added to the asphalt can increase the asphalt's weight and molecular density, affecting the test ball's friction resistance to be more significant [24]. The increase in weight and molecular density of asphalt, the amount of resin that increases along with the increase in crepe rubber content, causes an increase in the cohesion of the asphalt, which is characterized by the physical properties of the asphalt, namely the more sticky [25]. A significant decrease in softening point occurred in the sample with the addition of 3% TMQ (samples A3, B3, C3). The softening point reduction was due to the higher penetration of asphalt than asphalt pen grade 60/70 and other samples. Too high penetration indicates that the asphalt's cohesion properties will decrease, and it cannot hold the ball when testing the soft point. Increasing the asphalt's softening point can increase the asphalt resistance to rutting, increase ductility, thermal stability, and increase the aging time.

### 3.3. Effects of Crepe Rubber and TMQ on Short Term Aging of Modified Asphalt

The damage to the asphalt pavement during its service life is mostly due to the aging asphalt binder. Asphalt aging contributes to decreased durability and service life of asphalt pavements. Therefore, we need a testing method using the rolling thin film oven test (RTFOT) to simulate short-term aging on asphalt. The parameters for testing short term aging in this study used an approach to measure the weight loss of asphalt during RTFOT testing. Loss of weight in asphalt can occur due to volatile compounds such as aromatic components in asphalt [17]. Testing using RTFOT aims to determine volatile compounds' effect on short-term asphalt aging and asphalt's sensitivity to high temperatures. Meanwhile, long-term aging is caused by traffic loads, temperature, sunlight, oxidation, and others [26]. Loss of weight of asphalt allowed is a maximum of 1% of asphalt weight [8].

Sample B2 in Figure 3 experienced a weight loss of 0.059% during RTFOT testing or decreased by 83.84% compared to asphalt pen grade 60/70. Overall, sample B experienced constant weight loss compared to samples A and C. The addition of 10% crepe rubber in asphalt has the optimum condition in preventing the evaporation of volatile compounds or the crepe rubber part from evaporating. The average weight loss of asphalt in sample B is only 0.061%, while sample A has an average weight loss of 0.086% and 0.105% for sample C. as seen in Table 4. Logically the higher the rubber ratio, the smaller the weight loss that occurs after RTFOT is due to the function of crepe rubber in asphalt as a membrane that inhibits volatile compounds from evaporating [8]. However, this study concluded that the greater the ratio of crepe rubber to asphalt, the greater the weight loss. The weight loss of the asphalt occurs due to the crepe rubber's degradation during the RTFOT process, which results in the loss of some of the crepe rubber mass [2].

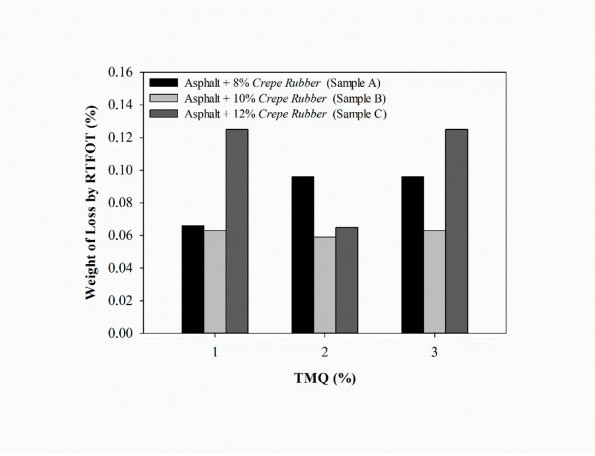

**Figure 3.** Weight of loss by RTFOT.

The loss of light components in the asphalt can reduce the penetration value and increase the softening point, as seen in Figure 4A. The penetration of sample B1 after RTFOT decreased until it reached a penetration value of 45.9 dmm. All samples experienced a decrease in penetration between 10–29% from before RTFOT. The low penetration of sample B1 causes the physical properties of sample B1 after penetration to become very hard and stiff. The decrease in penetration after RTFOT was due to several factors, including the asphaltene fraction, which increased along with the evaporation of the maltene fraction and rubber degradation, which increased in the carbonyl group. These two factors are the formation of free radicals by excess heat, which is based on age in the asphalt binding network [2].

Concerning Figure 4A,B presents data on the percentage reduction in asphalt penetration. In Figure 4B and Table 4, it can be seen that the addition of crepe rubber to the asphalt increases the percentage reduction in asphalt penetration. A significant decrease in asphalt penetration after RTFOT will increase the asphalt strength and make the asphalt susceptible to premature aging [27]. The increase in the percentage decrease in penetration of modified asphalt after RTFOT was caused by the effect of degradation of crepe rubber compared to the loss of light fraction in the asphalt. Degradation of crepe rubber plays a significant role in oxidation and increases in carbonyl groups, causing an increase in the proportion of asphaltene and a decrease in maltene [2]. In this case, crepe rubber accelerates the asphalt aging after RTFOT because crepe rubber is very susceptible to heat, oxidation, and UV. Antioxidants are necessary to prevent crepe rubber oxidation, which increases the risk of premature aging of the asphalt. This study uses TMQ as an antioxidant. The TMQ ratio's effect on the percentage reduction in asphalt penetration after RTFOT can be seen in Figure 4B. This study's optimal TMQ level was 2%, with a decrease in penetration after RTFOT of only 10–15%. TMQ can prevent rubber oxidation in 2 ways: stabilizing free radicals or scavenging free radicals and the second by inhibiting peroxide formation [2].

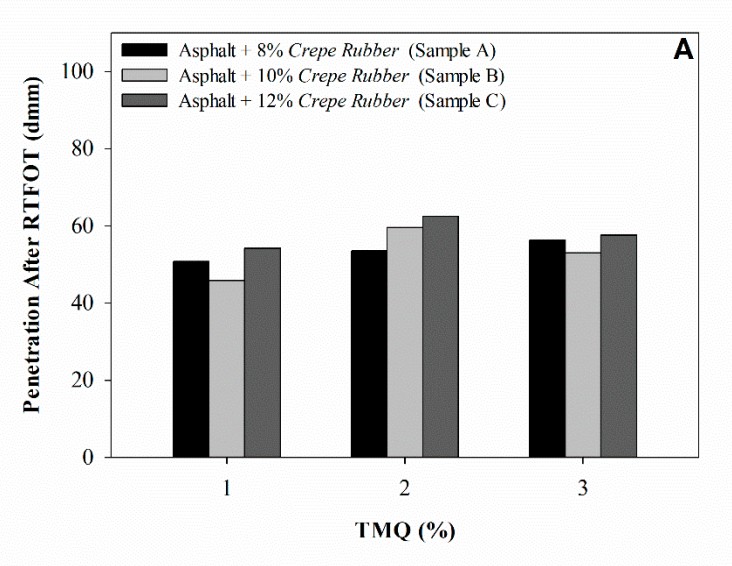

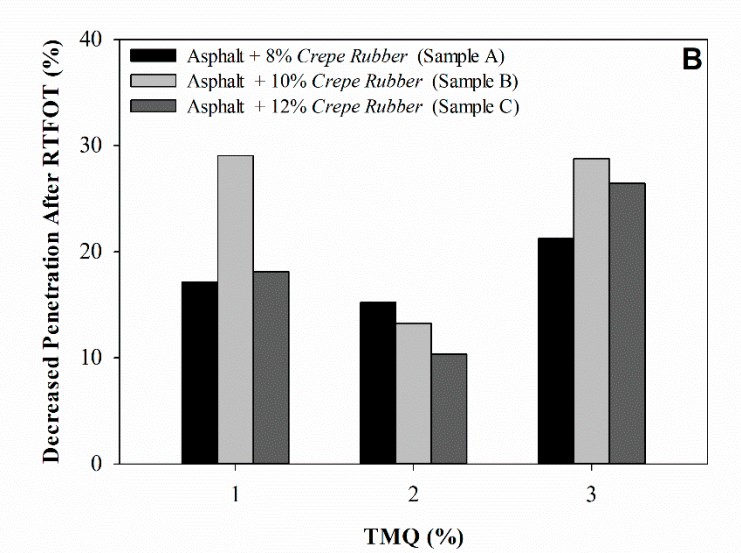

**Figure 4.** (**A**) Penetration after RTFOT. (**B**) Percentage of decreased penetration after RTFOT.

### 3.4. Effects of Crepe Rubber and TMQ on Characteristics of Modified Marshall Asphalt

Modified crepe rubber asphalt in its application for road pavement must have the ability to withstand traffic loads better than asphalt pen grade 60/70 [28]. Measuring modified asphalt to withstand road traffic loads can be done by laboratory-scale testing with Marshall stability testing. The Marshall stability value provides an overview of the asphalt's ability to bind aggregates and prevent deformation. The modified asphalt criteria can be successful if the Marshall stability value of asphalt with crepe rubber is higher than asphalt pen grade 60/70. The addition of crepe rubber to asphalt in its application for road pavement has advantages such as better riding quality, resistance to moisture, higher durability, and rutting due to reduced temperature sensitivity and lower noise levels.

Figure 5 shows the increase in the Marshall stability value in samples A1, A2, A3, B1, and B2 compared to pen grade 60/70 asphalt. The highest Marshall stability in samples A2 and B2 with a value of 1426.53 kg and 1403.96 kg or 20–21% of the 60/70 pen grade asphalt. In this study, the optimal asphalt content (KAO) was 5.45% for sample A and 5.50% for sample B with an aggregate composition, as shown in Table 2. Samples B3, C1, C2, and C3 experienced a decrease in the Marshall stability value

of asphalt pen grade 60/70 due to the penetration of the asphalt in the sample, which approaches and exceeds the asphalt pen grade 60/70. The shift in the penetration value will cause a change in the optimum asphalt content for mixing. The suboptimal bitumen content in the mixture affects the cohesion and adhesion properties of the asphalt. Inappropriate bitumen optimum content will cause the asphalt to be unable to bind the aggregate and easily deform or rutting permanently [29,30].

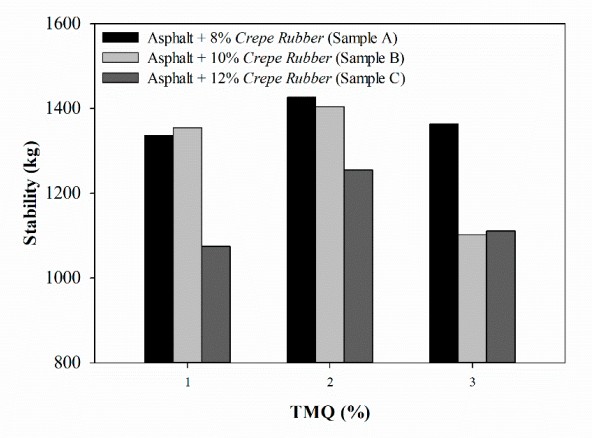

**Figure 5.** Marshall stability.

In general, crepe rubber's addition to the modified asphalt mixture can harden the asphalt binding with aggregate and increase Marshall stability. However, the addition of crepe rubber that exceeds the optimum will cause the asphalt to be more flexible and create instability when given a load [31]. The addition of TMQ helps strengthen the binding of asphalt with aggregate because TMQ only acts to prevent modified asphalt oxidation. The ability of modified asphalt to fill the aggregate cavity (VFA) affects the value of Marshall stability. The compacted asphalt mixture's cavities significantly affect the asphalt's stability, as shown in Figure 6. The large cavities allow cracks, which lead to a decrease in the stability of the asphalt.

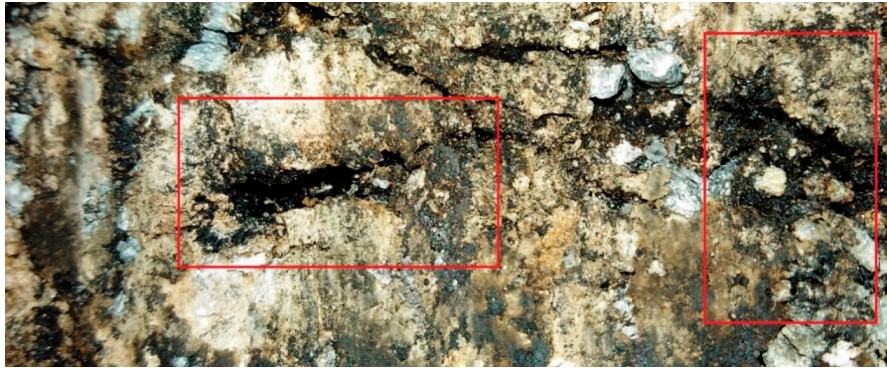

**Figure 6.** Cavities between aggregates in Marshall specimens.

In general, cavities that are too large after compaction will cause cracks during Marshall testing. Figure 6 is a piece of the Marshall sample B2 specimen. The red box shows that the large cavity between the aggregate grains causes cracks around the cavity after being loaded. Cracks that arise cause the resulting asphalt's stability is not as high as other research asphalt modifications. The value of cavities filled with asphalt or called void filled aggregate (VFA) shows the cavity's size in the asphalt.

VFA has a vital role in maintaining the stability of the mixture while holding loads. Table 5 and Figure 7A shows an increase in the percentage of VFA when the ratio of crepe rubber increases, but the VFA value decreases with 12% crepe rubber. The decreased VFA percentage in sample C

with 12% crepe rubber was due to the decrease in the optimum asphalt content, as shown in Table 4. The decrease in the optimum asphalt content affected the aggregate filler to fill the space between the aggregates. The asphalt level that does not reach the optimum can cause a cavity, as seen in Figure 6. The highest VFA value was obtained by sample B, with a 10% crepe rubber with an average VFA of 76.922%. The average percentage of VFA in sample C was higher than that of sample A, but the Marshall stability sample A was higher than that of Marshall stability sample C.

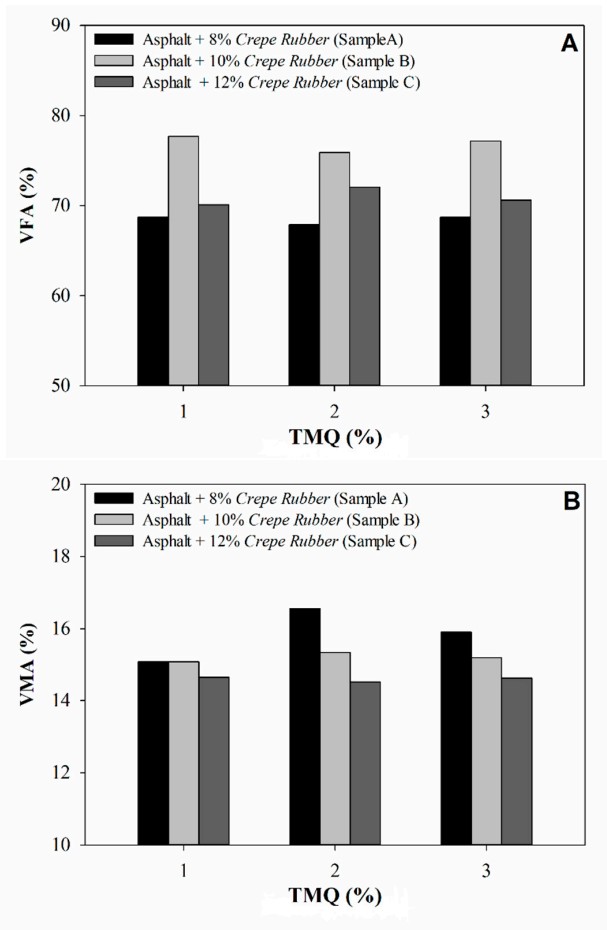

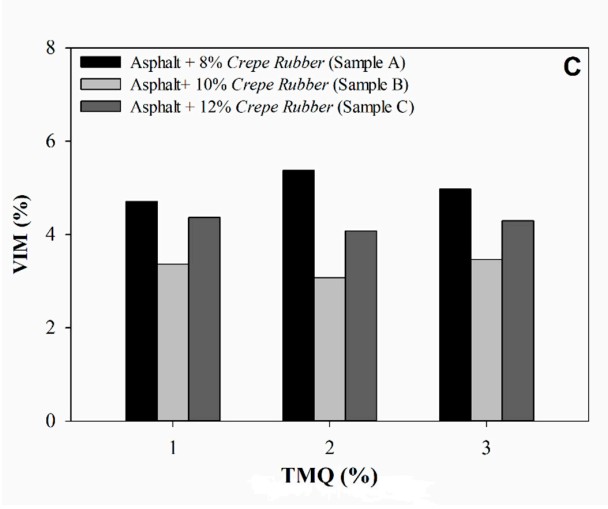

**Figure 7.** *Cont.*

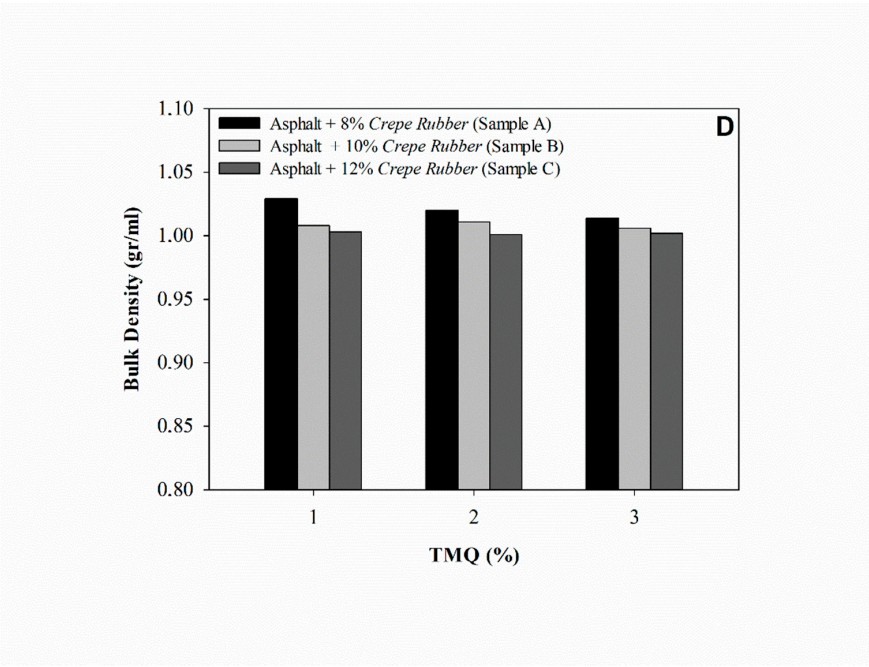

**Figure 7.** (**A**) VFA. (**B**) VMA. (**C**) VIM. (**D**) Bulk density modification asphalt.

The decrease in Marshall stability and the percentage VFA of sample C was due to several factors, including a smaller portion of optimum asphalt content. The optimum bitumen content, which is too small, will make the asphalt layer covering the aggregate thin and decrease the asphalt's adhesion properties [32]. The addition of crepe rubber and TMQ to the asphalt provides a significant change in modified asphalt density. As shown in Figure 7D, increasing the ratio of crepe rubber and TMQ changes the density of modified asphalt decreases, decreasing density is the cause of decreasing VFA percentage [33]. The increase in crepe rubber and excessive TMQ also creates instability because the binding power of asphalt with aggregates is higher than that of rubber. Combining these three things causes sample C to be no better than sample A in load-bearing stability. The percentage of VFA is closely related to the percentage of Void in mixture/air void (VIM) and void mineral aggregate (VMA). Increasing the percentage of VFA and decreasing the VMA value plays a vital role in the asphalt mixture's mechanical properties. In Figure 7B and Table 5, the increase in crepe rubber causes a decrease in VMA, the smallest VMA percentage in sample C with an average of 14.6%. The VMA percentage is too small; the mixture can experience durability problems because it forms a thin layer on the aggregate. Too large a percentage of VMA is also not good for asphalt mixtures because the stability will be very low [32].

Void in mixture/Air void (VIM) indicates the asphalt tightness and pore properties of asphalt. The percentage of VFA strongly influences VIM in the mixture. The small VFA percentage value indicates many voids between the asphalt aggregate mixture, as shown in Figure 8A. The VIM percentage represents the flexibility and durability of the asphalt and aggregate mixture. Asphalt with a VIM value that is too large indicates that the asphalt is not waterproof and is susceptible to oxidation and deformation when it is loaded. The VIM value that is too small also causes the mixture to become stiff and cracking due to the absence of cavities for the shifting aggregates. The smallest VIM value was obtained by sample B, with an average of 3.302%, as shown in Figure 7C and Table 5. The filler ratio has a significant effect on the percentage of VIM because filler has a vital role in filling the voids between aggregates. Additionally, the filler functions to make the mixture waterproof, as shown in Figure 8B.

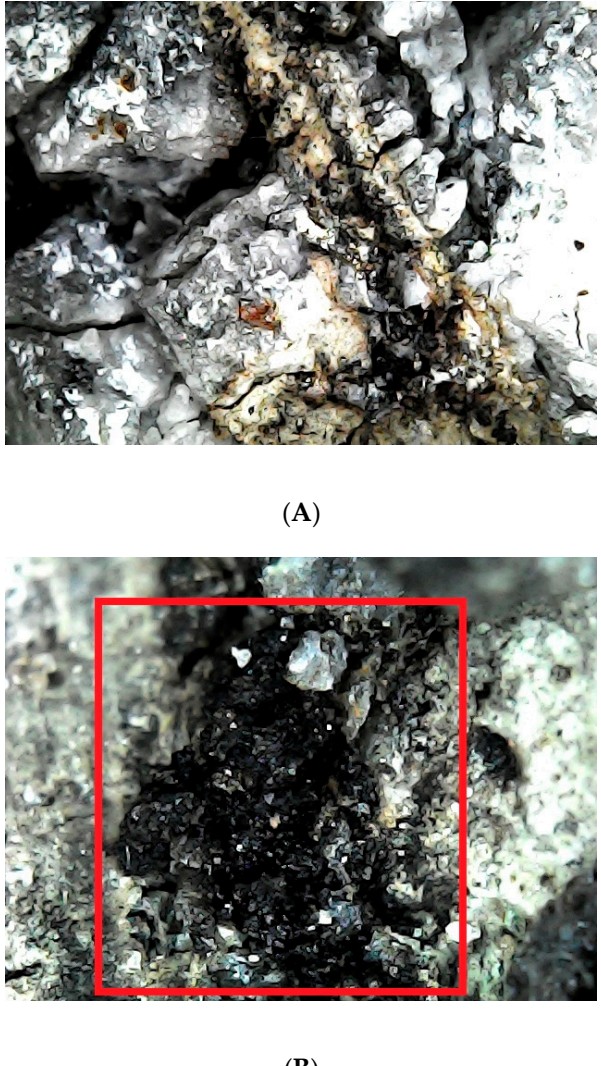

(**A**)

(**B**)

**Figure 8.** (**A**) Cavities that are not filled with modified asphalt. (**B**) Filler which fills the voids between the aggregates.

The asphalt mixture's stability can be influenced by several factors that significantly differ from other studies' results. The factors that affect the asphalt mixture's stability include the optimum asphalt content in the mix, the gradation and composition of the aggregate in the mix, the penetration of the asphalt-rubber used, the viscosity of the mixture, and the mixing and compaction temperature [34]. Increased bitumen stability provides better asphalt durability in withstanding traffic loads, increases rutting resistance, reduces deformation at high temperatures, reduces fatigue resistance, and prevents premature aging.

Based on the modified crepe rubber asphalt's characterization results, it was found that the asphalt with the addition of 10% crepe rubber and 2% TMQ (sample B2) had the most optimum conditions. Therefore, sample B2 was subjected to further testing to determine the asphalt performance after short-term aging and long-term aging.

*3.5. Modified Asphalt Performance at High Temperature before and after RTFOT*

3.5.1. Rutting Factor (G*/Sinδ)

The value (G*/Sinδ) is the quotient of the modulus complex (G*) and phase angle (δ), which can describe the performance of asphalt against rutting at high temperatures. Increasing the modulus

complex (G*) and decreasing the phase angle (δ) is desirable to produce asphalt with high rutting resistance. The higher the value (G*/Sinδ) shows the performance of asphalt in resisting rutting (anti-rutting) at high temperatures is better [35]. Figure 9 shows the values (G*/Sinδ) before and after RTFOT of asphalt pen grade 60/70, asphalt with 10% crepe rubber (B2 non-TMQ), and B2.

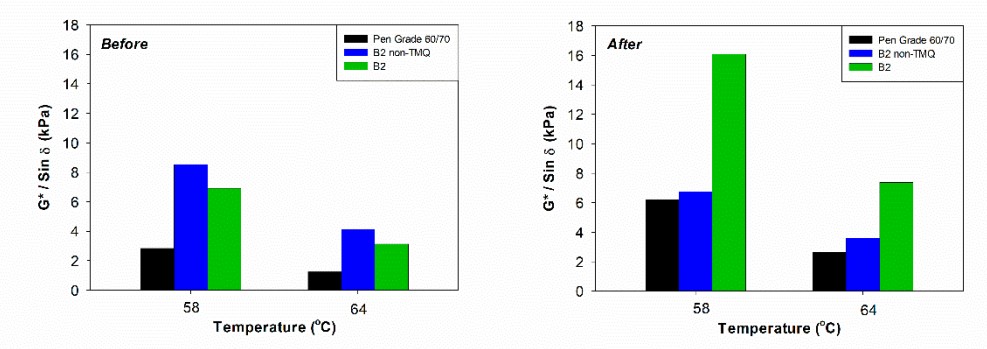

**Figure 9.** Factor rutting before and after RTFOT.

Figure 9 shows an increase (G*/Sinδ) after adding 10% crepe rubber and 2% TMQ (Sample B2 non-TMQ and B2). However, it decreases with increasing test temperature because the asphalt is viscous. The increase in value (G*/Sinδ) in the B2 non-TMQ and B2 samples indicates that the asphalt properties are getting harder after the addition of crepe rubber. The highest value (G*/Sinδ) reached 6.91 kPa before RTFOT, while asphalt pen grade 60/70 was only 2.86 kPa, and B2 non-TMQ was only 5.46 kPa. This is closely related to the decreased penetration value of asphalt after the addition of 10% crepe rubber and 2% TMQ compared to asphalt pen grade 60/70. This is due to the increase in cohesion and adhesion and the bitumen's resistance to low temperatures, marked by an increase in the softening point value (see Figure 2). In this case, crepe rubber plays a role in increasing the cohesion and adhesion of the asphalt. One of the factors affecting bitumen's resistance to rutting and fatigue is the modified asphalt's cohesion and adhesion properties. The addition of crepe rubber increases the asphalt mixture with aggregates to increase asphalt cohesion and adhesion performance.

However, the addition of TMQ (sample B2) shows that the rutting factor decreases before the TFOT process is carried out (see Figure 9). This is because the penetration of asphalt increases as the concentration of TMQ in the mixture increases. The increase in penetration indicates that the asphalt's cohesion and adhesion properties decreased due to the antioxidant TMQ, which melts during mixing. TMQ antioxidant has physical properties such as wax, causing the asphalt's viscosity to decrease, and the fraction of maltene in the asphalt increases after mixing.

The value of the rutting factor (G*/Sinδ) of the 60/70 pen grade asphalt, B2 non-TMQ, and B2 asphalt samples increased with aging during RTFOT and decreased with increasing temperature. The value (G*/Sinδ) after RTFOT has a good appearance with a decrease in the value of asphalt penetration after RFTOT. This indicates that it is getting louder after RTFOT. The value (G*/Sinδ) increases drastically in sample B2 due to the higher elastic proportion after RTFOT. The increase in the rutting factor in sample B2 after RTFOT showed that the cohesion and adhesion of asphalt increased rapidly due to TMQ's performance stabilizing the rubber phase on the asphalt and preventing aging. This result correlates with the increase in Marshall stability, where asphalt with the addition of crepe rubber and 2% TMQ (sample B2) has a high stability value (see Figure 5). High stability indicates that the sample has high rutting and fatigue resistance. However, a significant increase in value (G*/Sinδ) is not expected as this indicates the bitumen is too stiff and decreases the elastic properties of the asphalt. Asphalt with good rutting resistance must be rigid enough and elastic enough to return to its original shape after loading [36].

The DSR test results of the asphalt B2 non TMQ and B2 samples showed that the asphalt performance after rubber was better than that of asphalt pen grade 60/70. Sample B2 had the best



rutting resistance performance at elevated temperatures before RTFOT. However, after the RTFOT, there was a significant increase in value ($G^*/Sin\delta$) to 16.1 kPa. Based on the rutting factor data described, the results of this test have a reasonable correlation with the asphalt Marshall's stability. A high Marshall stability value indicates that the bitumen has better cohesion and adhesion properties and rutting resistance.

### 3.5.2. Phase Angle (δ)

The phase angle (δ) shows the strain versus stress results in the viscoelasticity properties of asphalt. The phase angle reflects the proportion between the elasticity and viscous in the asphalt. The phase angle value can be 0 (δ = 0°) if the material is fully elastic and has a value of 90° (δ = 90°) if the material is fully viscous [35,37]. However, if the material is viscoelastic, the phase angle value will be 0°–90°. Figure 10 shows the asphalt pen grade 60/70 samples' phase angles, B2 non-TMQ, and B2 before and after RTFOT.

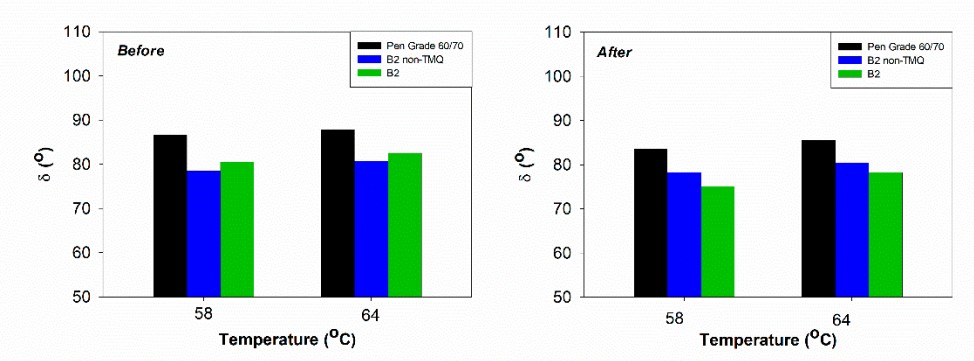

**Figure 10.** Phase angle before and after RTFOT.

Figure 10 shows that the phase angle decreases with the addition of crepe rubber and TMQ both before and after aging. The decrease in phase angle after the addition of crepe rubber and TMQ (Sample B2 non-TMQ and B2) was due to the increase in the elastic proportion after crepe rubber and TMQ. At 58 °C, the phase angle of asphalt pen grade 60/70 was 86.6°, while the non-TMQ and B2 samples had 81.4° and 80.5° phase angles before RTFOT. However, the phase angle of all samples increases with increasing temperature.

After RTFOT, all samples experienced a decrease in phase angle and increased with increasing temperature. The reduction in phase angle after aging is due to the increased proportion of elasticity after RTFOT. This has a good correlation with decreasing asphalt penetration after RTFOT. Visually, the decrease in phase angle is due to the stronger asphalt binder network, indicating that the asphalt is getting harder. In general, the reduction in phase angle suggests that the addition of crepe rubber and TMQ plays a vital role in the viscoelasticity of the asphalt [38].

The phase angle of sample B2 decreased due to the addition of TMQ, which increased the elastic proportion in the asphalt-rubber mixture. Comparing the phase angles of the asphalt pen grade 60/70 samples. B2 non-TMQ, and B2, the phase angle increases with crepe rubber and TMQ. The addition of crepe rubber increases the elastic proportion, while the increase in temperature affects the viscous ratio.

### 3.5.3. Complex Modulus G*

The complex modulus ($G^*$) is a parameter to indicate the asphalt's binder strength (the cohesion and adhesion properties of asphalt), which determines the resistance to deformation. The combination of elastic and viscous shear modulus reflects the bitumen's resistance to deformation, which is

influenced by the elastic proportion and viscosity. Therefore, the complex modulus (G*) is a factor that needs to be considered in assessing the performance of asphalt against high temperatures and aging.

Figure 11 complex modulus (G*) of asphalt samples pen grade 60/70. B2 non-TMQ, and B2 in the conditions before and after RTFOT. The increase in the complex modulus value occurred in the non-TMQ B2 and B2 samples compared to asphalt pen grade 60/70. The increase in the complex modulus (G*) value obtained by adding crepe rubber increases the asphalt's elastic proportion. However, the complex modulus of sample B2 was much higher after the addition of TMQ, which further increased the rubber-asphalt elasticity.

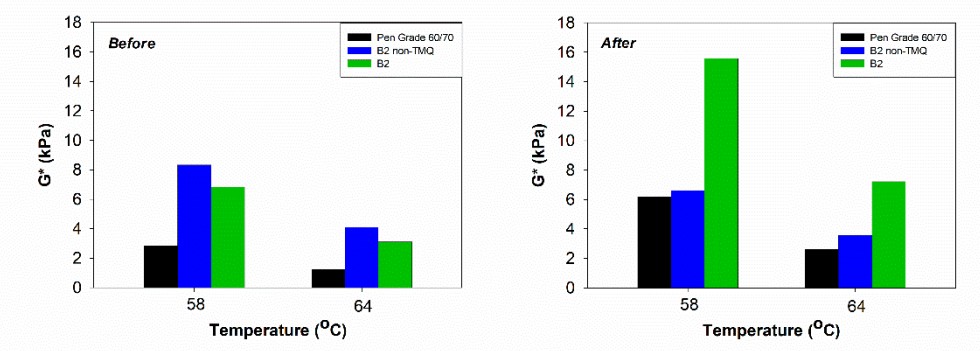

**Figure 11.** Complex Modulus Before and After Aging (RTFOT).

In all samples, the complex modulus values increase with aging and decrease with increasing temperature. The elastic proportion influences the increase in the complex modulus value after RTFOT. Complex modulus (G*) results after RTFOT have a reasonable correlation with decreasing penetration after RTFOT. The penetration that decreased after RTFOT indicated that the asphalt binder was getting harder. Sample B2 had a very significant increase in the complex modulus value after RTFOT from 6.91 kPa to 16.1 kPa. The increasing elastic proportion influences this significant increase because asphalt loses some light fractions during RTFOT. These modulus complex results are consistent with the rutting and phase angle results, where an increase in other components increases the elastic proportion. In contrast, an increase in temperature changes the viscous proportion to an increase [39,40].

### 3.6. FT-IR Modified Asphalt before and after LTOA

The performance of crepe rubber and TMQ antioxidants against long-term aging of asphalt can be simulated using the Long Term Oven Aging (LTOA) method [41]. Furthermore, long-term characterization using FTIR analysis in the range of wavenumbers from 4000 to 600 cm$^{-1}$. Long-term asphalt aging is indicated by the formation of carbonyl groups (-C=O) at the wave number 1700 cm$^{-1}$ and sulfoxide (-S=O) at the wave number 1030–1070 cm$^{-1}$ [42,43].

Figure 12A illustrates the results of FTIR analysis for samples of asphalt pen grade 60/70. asphalt with the addition of 10% crepe rubber (B2 non-TMQ) asphalt with the addition of crepe rubber and 2% TMQ (B2) before aging. The results showed that there were several absorption peaks at several wavelengths including, 3670 cm$^{-1}$ (Free OH), 2930 cm$^{-1}$ (CH2, CH3), 1600 cm$^{-1}$ (C=C), 1455 cm$^{-1}$ (CH3 Deformation), 1379 cm$^{-1}$ (CH3 Deformation), 1060 cm$^{-1}$ (SO or CC aromatic (crepe rubber or TMQ)), 825 cm$^{-1}$, 745 cm$^{-1}$ (CH aromatic) [44,45]. The addition of crepe rubber to the asphalt did not chemically affect the asphalt. This was evidenced by the FTIR results graph, which did not find any new clusters. However, with asphalt with crepe rubber and TMQ, there was an increase and decrease in the number of clusters that were not so specific. At wavenumbers 3670 and 1060 cm$^{-1}$, there is an increase in the hydroxy group not attached to hydrogen and generally found in the form of alcohol or phenol with the OH group sterically inhibited. It should be noted that the wavenumber 3670 cm$^{-1}$ can also indicate free OH groups and free from interaction with ions or other groups [45].

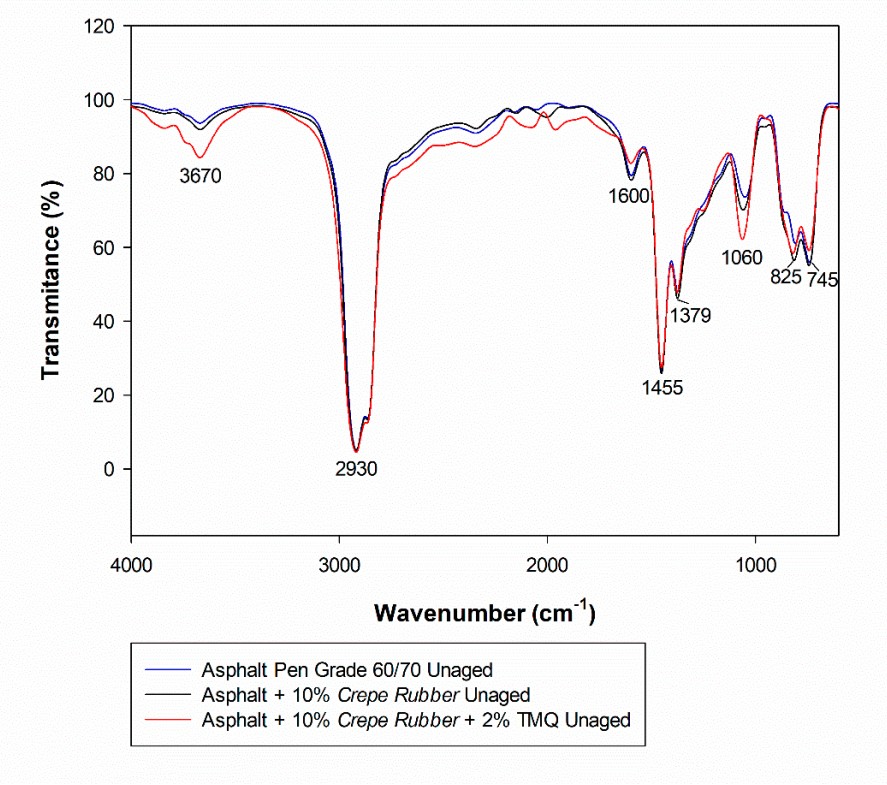

(**A**)

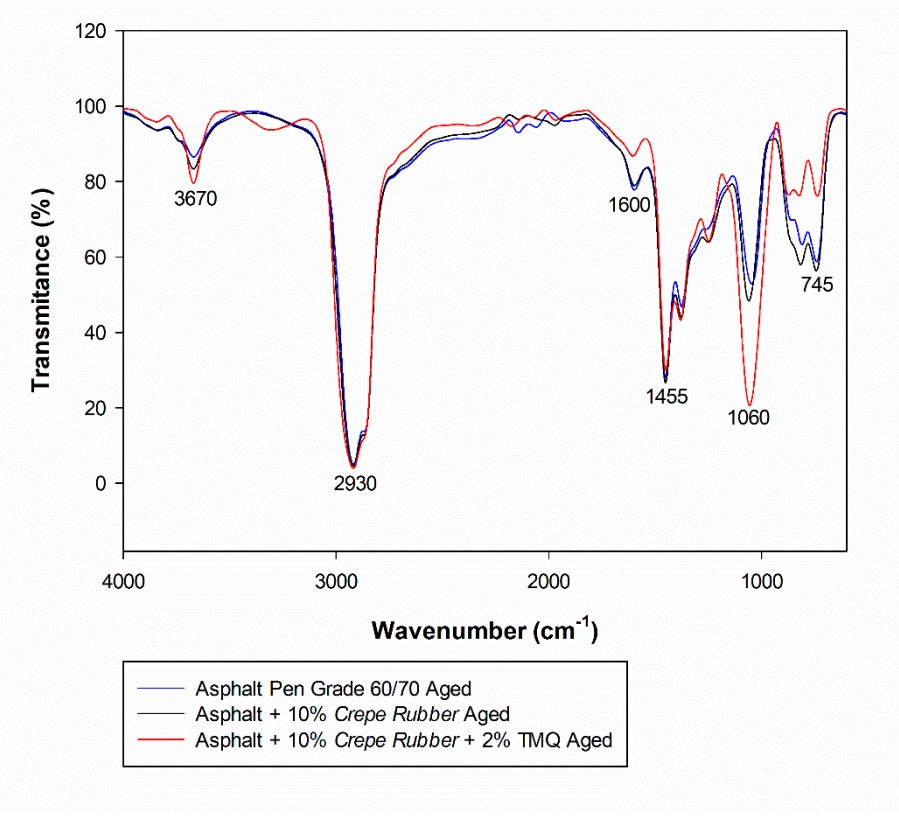

(**B**)

**Figure 12.** (**A**) FT-IR of asphalt before LTOA. (**B**) FT-IR asphalt after LTOA.

While the group at wave number 1060 cm$^{-1}$ shows the sulfoxide (S-O) groups of asphalt [42–44]. Naturally, asphalt has a sulfoxide group because asphalt contains a certain amount of sulfur. Due to storage for a long time, it causes oxidation of the asphalt to form sulfoxide groups. But the addition of crepe rubber and TMQ antioxidants in the asphalt mixture increased the number of groups at the wavenumber of 1060 cm$^{-1}$. The increase in these groups was not due to an increase in sulfoxide groups but increased the aromatic C-C group from crepe rubber and TMQ [46].

Figure 12B. It is an illustration of FTIR analysis results for samples of asphalt pen grade 60/70, asphalt with the addition of 10% crepe rubber, and asphalt with the addition of crepe rubber and 2% TMQ after aging. By comparing Figure 12A,B, it is not found to include new groups in the sample, especially the carbonyl group (1700 cm$^{-1}$), indicating aging on the asphalt. The indication of asphalt aging is shown in the increase of sulfoxide groups after the aging process peaked at 1060 cm$^{-1}$. However, at the same wavenumber, it can be seen that the sample with the addition of crepe rubber and TMQ has an increasingly steeper peak. The peak that is formed is getting steeper, indicating an increase in the concentration of a group. In this case, the aromatic C-C group's increase played a significant role in the 1060 cm$^{-1}$ peak's steepening.

The increase in aromatic C-C groups tends to be caused by the degradation of crepe rubber or TMQ during the aging process [46]. Aging of asphalt samples with a mixture of crepe rubber and TMQ showed decreased peak sharpness at wavenumbers 825 cm$^{-1}$ and 745 cm$^{-1}$ (C-H aromatic). The decrease in peaks of 825 cm$^{-1}$ and 745 cm$^{-1}$ showed the performance of TMQ, which suppressed carbonyl group formation by changing the aromatic C-H groups, which were the source of radical formation to become aromatic C-C. The increase in the carbonyl group should be directly proportional to the increase in aromatic C-H [47]. In general, the role of TMQ in overcoming oxidation in the rubber asphalt mixture is due to the absence of carbonyl groups in all asphalt samples after aging using the LTOA method. This is consistent with references where TMQ plays a vital role in preventing polymers [46,48].

## 4. Conclusions

The use of crepe rubber as an asphalt additive can improve the performance of asphalt pen grade 60/70. The most noticeable improvements were load-bearing capabilities and thermal stability. Modified asphalt has lower penetration, soft point, and high Marshall stability than pen grade 60/70 asphalt. The addition of 10% crepe rubber and 2% TMQ is the most optimum condition with the following test results: penetration 68.70 dmm, softening point 55.45 °C, weight loss only 0.0579%, penetration after RTFOT 59.60 dmm. Marshall stability 1403.96 kg with an optimum asphalt content of 5.50% and a rutting factor (G*/Sin) of 6.91 kPa and 16.1 kPa before and after RTFOT. Increased bitumen stability provides better asphalt durability in withstanding traffic loads, increases rutting resistance, reduces deformation at high temperatures, reduces fatigue resistance, and prevents premature aging. The addition of antioxidant TMQ more than 2% changed the asphalt's physical properties to become softer, increased penetration, and decreased Marshall stability. The performance of modified asphalt 10% crepe rubber and 2% TMQ after RTFOT increased after the addition of TMQ. The most obvious increase was the rutting factor (G*/Sinδ) from 6.91 kPa to 16.1 kPa after RTFOT. TMQ performance is outstanding under long-term aging conditions, where the carbonyl group is not formed (C = O) after the addition of TMQ seen in the FT-IR results.

**Author Contributions:** Conceptualization, B.I.; Formal analysis, B.I. and A.W.; Investigation, A.M.; Methodology, B.I.; Validation, A.M.; Visualization, A.W. All authors have read and agreed to the published version of the manuscript.

**Funding:** This research was funded by DRPM Kemristek/BRIN, the Republic of Indonesia.

**Conflicts of Interest:** The authors declare no conflict of interest.

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
