# Peer review of "The Effect of Addition of Antioxidant 1,2-dihydro-2,2,4-trimethyl-quinoline on Characteristics of Crepe Rubber Modified Asphalt in Short Term Aging and Long Term Aging Conditions"

_applsci, doi:10.3390/app10207236_

Round 1
Reviewer 1 Report
- Paper needs significant editing to improve readability and technical writing. For example, rewrite paras 1 and 2 on page 3. The content is not clear for the reader.
- What is "viscosity mooney" on page 3, second last para?
- Check English on page 5 in the last para.
- How many replicates were used for lab tests? Only averages are shown in the figures to conclude on the effects. It will be more scientific to include statistical analysis of the data to consider variability with replicates.
- How can the addition of additives improve both rutting and fatigue resistance of the HMA as mentioned in the conclusions? More discussion is needed on this aspect.
- Consider reducing the paper length by eliminating redundant materials. The focus o the paper should be only on the effect of additives on aging characteristics.
Author Response
Thank you for reviewing the manuscript entitled "The Effect of Addition of Antioxidant 1,2-dihydro-2,2,4-trimethyl-quinoline on Characteristics of Crepe Rubber Modified Asphalt in Short Term Aging and Long Term Aging Conditions"
We have made some improvements to the manuscript according to the suggestions of reviewers. We respond to reviewers' questions in the form of the attached file.
Please see the attachment
Reviewer 2 Report
Dear authors,
please find comments in attach.
Regards

Author Response
Dear Reviewer,
Thank you for reviewing the manuscript entitled "The Effect of Addition of Antioxidant 1,2-dihydro-2,2,4-trimethyl-quinoline on Characteristics of Crepe Rubber Modified Asphalt in Short Term Aging and Long Term Aging Conditions"
We have made some improvements to the manuscript according to the suggestions of reviewers. We respond to reviewers' questions in the form of the attached file.
Please see the attachment
Regards

Round 2
Reviewer 1 Report
Did you conduct any statistical analysis to compare different means?Author Response
Dear reviewer,
In the manuscript that we have sent, we have not discussed statistical analysis for the data displayed in the manuscript. Perhaps for the next publication, we will complement the results and discussion by presenting a statistical analysis of the data obtained.
Regards